# Engineered CRISPR-OsCas12f1 and RhCas12f1 with robust activities and expanded target range for genome editing

Xiangfeng Kong[1,2,5], Hainan Zhang[1,2,5], Guoling Li[2,5], Zikang Wang[1,2,5], Xuqiang Kong[1,2,5], Lecong Wang[1,2], Mingxing Xue[1,2], Weihong Zhang[1,2], Yao Wang[1,2], Jiajia Lin[3], Jingxing Zhou[1,2], Xiaowen Shen[1,2], Yinghui Wei[1,2], Na Zhong[1,2], Weiya Bai[2], Yuan Yuan[2], Linyu Shi[2], Yingsi Zhou[1,2] ✉ & Hui Yang [1,2,4] ✉

The type V-F CRISPR-Cas12f system is a strong candidate for therapeutic applications due to the compact size of the Cas12f proteins. In this work, we identify six uncharacterized Cas12f1 proteins with nuclease activity in mammalian cells from assembled bacterial genomes. Among them, OsCas12f1 (433 aa) from *Oscillibacter* sp. and RhCas12f1 (415 aa) from *Ruminiclostridium herbifermentans*, which respectively target 5' T-rich Protospacer Adjacent Motifs (PAMs) and 5' C-rich PAMs, show the highest editing activity. Through protein and sgRNA engineering, we generate enhanced OsCas12f1 (enOsCas12f1) and enRhCas12f1 variants, with 5'-TTN and 5'-CCD (D = not C) PAMs respectively, exhibiting much higher editing efficiency and broader PAMs, compared with the engineered variant Un1Cas12f1 (Un1Cas12f1_ge4.1). Furthermore, by fusing the destabilized domain with enOsCas12f1, we generate inducible-enOsCas12f1 and demonstate its activity in vivo by single adeno-associated virus delivery. Finally, dead enOsCas12f1-based epigenetic editing and gene activation can also be achieved in mammalian cells. This study thus provides compact gene editing tools for basic research with remarkable promise for therapeutic applications.

The facile, RNA-programmable, CRISPR-Cas system, which serves as an adaptive immune system against phage infection and foreign plasmids in bacteria, has been developed into a versatile tool for genome editing and modifying the regulation of gene expression in various organisms[1–3]. CRISPR-Cas systems are divided into two classes (class 1 and 2) and six major types (type I–VI), with >30 subtypes, based on the diversity of Cas effectors as well as the architecture of their genomic loci[4]. The class 2 system, which includes CRISPR-Cas9[5,6], -Cas12a[7], and their derived gene editing tools[8–12], are widely used for basic research, gene therapy, and development of agricultural biotechnology[1,2,13]. Nevertheless, their large size, which is typically >1000 amino acids (aa), surpassing the packaging limit in viral vectors, which consequently hinders their delivery.

Recently, an exceptionally compact class 2 type V-F CRISPR-Cas system that uses a Cas12f effector protein (400–700 aa) had been reported functional in eukaryotes[14–17]. By engineering the sgRNA, the

[1]HUIEDIT Therapeutics Co., Ltd., Shanghai 200131, China. [2]HUIDAGENE Therapeutics Co., Ltd., Shanghai 200131, China. [3]Department of Neurology, First Affiliated Hospital, Fujian Medical University, Fuzhou, China. [4]Institute of Neuroscience, State Key Laboratory of Neuroscience, Key Laboratory of Primate Neurobiology, Center for Excellence in Brain Science and Intelligence Technology, Chinese Academy of Sciences, Shanghai 200031, China. [5]These authors contributed equally: Xiangfeng Kong, Hainan Zhang, Guoling Li, Zikang Wang, Xuqiang Kong. ✉e-mail: yingsizhou@huidagene.com; huiyang@huidagene.com

recombinant Un1Cas12f1_ge4.1 variant, which can be packaged into a single recombinant adeno-associated virus (rAAV) vector, was found to exhibit high-editing efficiency (i.e., comparable to that of SpCas9) at some genomic loci[15]. This finding suggested considerable potential for adoption of type V-F CRISPR-Cas systems for therapeutic editing in vivo. However, the restricted 5'-TTTR PAM of Un1Cas12f1_ge4.1 may hinder its broad application. The application of therapeutic genome editing, especially for genomic alterations that require systemic delivery, could be significantly advanced by development of compact, broad target range, high efficiency, and high fidelity Cas12fs that are packageable in single rAAV capsid vectors. In addition, controllable gene editing system in vivo is always of great interest to the field to reduce the concern of off-target effects.

Here, we present two hypercompact Cas12f1s from *Oscillibacter* sp. (OsCas12f1) and *Ruminiclostridium herbifermentans* (RhCas12f1). Through protein engineering and sgRNA optimization, we generate the enhanced OsCas12f1 (enOsCas12f1) and enRhCas12f1 variants, showing higher editing efficiency with low off-target effects, as well as a wider range of target loci recognition in human cells. Furthermore, enOsCas12f1 and its inducible version are applied for efficient restoration of dystrophin in humanized mdx mice by single AAV

delivery. Additionally, enOsCas12f1 can be engineered for both epigenome editing and gene activation.

## Results

### Identification and characterization of type V-F CRISPR-Cas12f1 systems

In order to identify high-efficiency CRISPR-Cas12f1 systems with potential therapeutic application, we downloaded ~200,000 bacterial genomes from NCBI. We developed and employed a computational pipeline to annotate Cas12f1 orthologs, CRISPR array, tracrRNAs, and PAM preferences, which led to the identification of 34 previously uncharacterized CRISPR-Cas12f1 systems and their respective tracrR-NAs (Supplementary Data 1). Our criteria for identifying Cas12f1 candidates are: i) native gene organization similar to CRISPR-Cas12f1 system[4] (Supplementary Fig. 1a), ii) predicted tracrRNA shows secondary structure (Supplementary Fig. 1b), iii) tracrRNA contains anti-repeat region that is able to form base pairs with crRNA (Supplementary Fig. 1c, d). These systems were then phylogenetically clustered into two subgroups based on Cas12f1 effector protein sequence alignment and putative PAM preferences (Fig. 1a). To evaluate the efficiency of dsDNA cleavage in eukaryotic cells by these CRISPR-

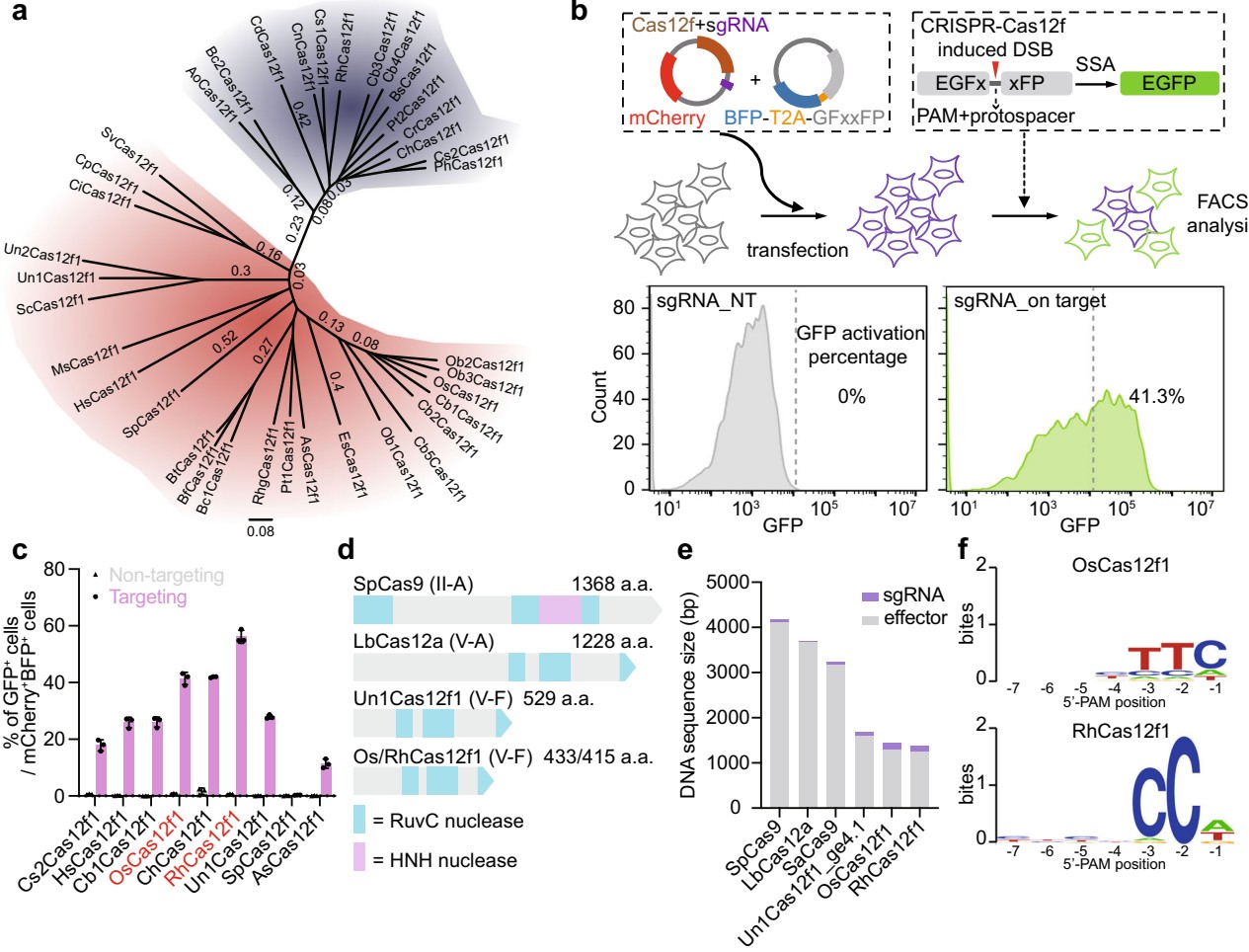

**Fig. 1 | Identification and characterization of CRISPR loci and Cas protein of type V-F system. a** Maximum-likelihood tree of identified Cas12f1 and previously reported Cas12f1. The evolutionary distance scale of 0.08 is shown. **b** Scheme of Cas12f1-induced EGFP activation in HEK293T cells. Transfection of plasmids expressing Cas12f1 and sgRNA activated EGFP. **c** Various Cas12f1 mediated EGFP activation efficiency determined by flow cytometry. Values and error bars represent mean and s.d. (*n* = 3). Target sequence: CCATTACAGTAGGAGCATAC. NT means sgRNA with random spacer sequence. The two most efficient Cas12f1s selected for

further study were highlighted in red. FACS gating strategy shown in Supplementary Fig. 1e. **d** Protein organization of SpCas9, LbCas12a, Un1Cas12f1_ge4.1, OsCas12f1, and RhCas12f1. Nuclease domains including RuvC and HNH, as well as protein length are indicated. **e**, Comparison of DNA sequence size of OsCas12f1, RhCas12f1, and other commonly used CRISPR system. **f** WebLogos of the PAM sequences for OsCas12f1 and RhCas12f1. Source data are provided as a Source Data file.

Cas12f1 systems, we designed an enhanced green fluorescent protein (EGFP) reporter system activated by single-strand annealing (SSA)-mediated repair pathway in HEK293T cells[18]. This system relied on co-transfection with two plasmids, the first of which expresses a BFP-T2A-EGFxxFP cassette, with a deactivated EGFP harboring a short insertion sequence between EGFx (EGFP CDS 1−561 bp) and xFP (EGFP CDS 112−720 bp) that is replaceable with endogenous PAM-containing sequence. The other plasmid carried an expression cassette for the amino- and carboxyl-terminal nuclear localization sequence (NLS)-tagged Cas12f1 and its sgRNA, which consists of a tracrRNA fused with a mature crRNA by a GAAA tetraloop, that targets the insertion sequence in the reporter plasmid (Fig. 1b and Supplementary Data 1). Cas12f1-triggered DSBs induce the SSA-mediated repair of EGFxxFP, consequently activating the EGFP (Fig. 1b and Supplementary Fig. 1e).

Using this screen, we functionally characterized six CRISPR-Cas12f1 systems (Fig. 1c and Supplementary Fig. 1f). Based on our observations of robust EGFP activation by OsCas12f1, HsCas12f1, Cb1Cas12f1, and RhCas12f1 in HEK293T cells, we next validated the frequency of indel generated by these Cas12f1s at endogenous genomic loci. The results showed that the genomic editing efficiencies of OsCas12f1, HsCas12f1, Cb1Cas12f1, and RhCas12f1 were modest, with indel frequencies ranging from 1% to 20% at various target loci (Supplementary Fig. 2).

We then selected the two CRISPR-Cas12f1 systems with highest GFP activation efficiency for further study, including OsCas12f1 (433 aa) and RhCas12f1 (415 aa), which recognize 5′ T- and 5′ C-rich PAMs, respectively. Both OsCas12f1 and RhCas12f1 are hypercompact, with a gene size that is less than half of SpCas9, LbCas12a, and SaCas9 (Fig. 1d, e). In vitro cleavage of a DNA fragment library containing 7-bp random sequence indicated that OsCas12f1 and RhCas12f1 recognized PAM of 5′-YTTH (Y = C or T, H = not G) and 5′-NCCD (D = not C), respectively (Fig. 1f). We explored the effects of spacer length on cleavage efficiency in OsCas12f1 and RhCas12f1, which showed that a 20nt spacer was optimal for their activation (Supplementary Fig. 3a, b). By introducing point mutations that resulted in D228A or D406A residue conversions in the conserved active sites of the RuvC domain, OsCas12f1 cleavage activity was abolished (Supplementary Fig. 3c, d). Similarly, RhCas12f1 could be inactivated through nonsynonymous point mutations leading to D210A or D388A conversion mutations (Supplementary Fig. 3e).

Next, we tested the biochemical properties of OsCas12f1 and RhCas12f1 proteins (Supplementary Fig. 4a). Linear plasmids cleavage assay suggested that both OsCas12f1 and RhCas12f1 were dsDNA cleavage active at a wide range of temperature, preferring 37 °C−50 °C (Supplementary Fig. 4b). The cleavage activities of OsCas12f1 and RhCas12f1 were validated on both supercoiled and linear plasmids (Supplementary Fig. 4c). To characterize the dsDNA cleavage pattern of OsCas12f1 and RhCas12f1, run-off sequencing of in vitro cleavage products were performed, indicating OsCas12f1 and RhCas12f1 cut dsDNA at sites of 21-25 bp downstream of the 5′-PAM with sticky ends (Supplementary Fig. 4d, e). Size-exclusion chromatography was performed to determine the complex formation of Cas12 f1 protein with its sgRNA, suggesting that both OsCas12f1 and RhCas12f1 could form dimer in presence of sgRNA at least in the tested condition, which was similar to that of Un1Cas12f1[19,20] (Supplementary Fig. 5).

Taken together, these results indicated that OsCas12f1 and RhCas12f1 offer hypercompact DNA editing tools with modest genomic editing efficiency and relatively wide target range.

## Arginine substitution in the REC/RuvC domains and C-G base pair replacement in the sgRNA enhanced cleavage efficiency of OsCas12f1 and RhCas12f1

In order to increase the cleavage efficiency of OsCas12f1 and RhCas12f1, we then sought to engineer these Cas12f1 proteins through mutagenesis and screening for higher efficiency variants using the

same GFP activation reporter system, as described above (Fig. 1b). Previous structural analysis and biochemical characterization of Un1Cas12f1[19,20], combined with reported strategies for increasing Cas12 activity[14,21,22], led us to speculate that substituting amino acids in the RNA or DNA recognition region for positively charged arginine (R) residues may increase Cas12f1 activity by facilitating its interactions with sgRNA and/or target DNA. Based on the protein alignment of OsCas12f1 and RhCas12f1 with Un1Cas12f1, three regions that potentially responding for binding nuclei acids were defined (Supplementary Fig. 6). Amino acids (except for positively charged residues including lysine, arginine, and histidine) in the region1-3 of OsCas12f1 and RhCas12f1 were individually mutated into arginine by mutagenesis method as previously reported[23] (Supplementary Fig. 7). We, therefore, generated two mutant libraries within these three regions, each respectively containing over 100 protein variants of OsCas12f1 and RhCas12f1. These variants were then individually co-transfected with the reporter plasmid into HEK293T cells and EGFP activation efficiency was quantified by flow cytometry (Fig. 2a). Although most variants showed similar or lower efficiency to that of wild-type OsCas12f1 (WTOsCas12f1), a subset of variants exhibited increased activity (Fig. 2b and Supplementary Table 1). The most efficient OsCas12f1 variant, D52R, showed 1.31-fold improvement over WTOsCas12f1 (Fig. 2b). To determine whether substitution with other amino acids could further enhance cleavage efficiency over that of the R substitution variants, we mutagenized D52 to saturation and found that R substitutions indeed conferred a better or slightly better OsCas12f1 nuclease activity (Supplementary Fig. 8a).

Second round iteration screen was performed by mutating OsCas12f1-D52R with additionally incorporating one more mutation that was identified as enhanced OsCas12f1 mutants in our first round screen. Using a library containing 15 double mutants of OsCas12f1, we found that R substitution at A54, S119, T132, and S141 further increased the activity of OsCas12f1-D52R (Fig. 2c). We thus selected the most efficient OsCas12f1 mutant containing T132R/D52R double mutation for further engineering.

Modification of the 3′ poly-uridine (U) overhang on gRNAs has been shown to increase gRNA stability, and consequently improve Cas nuclease efficiency[15,24,25]. In the current study, we subsequently fused a 5′-TTTTATTTTTTT-3′ sequence to the 3′ of sgRNAs and adopted an sgRNA optimization strategy similar to that used for Un1Cas12f1[14,15], including truncation or deletion of base pairs in the RNA stem region (Fig. 2d and Supplementary Data 2). However, the cleavage activity of OsCas12f1 was impaired rather than increased by truncation of the RNA stem (Fig. 2e). Alternatively, we next replaced the A-U or mismatched base pairs to thermodynamically stable C-G base pair which may increase sgRNA stability (Fig. 2d and Supplementary Data 2). These sgRNA variants resulted in substantially higher OsCas12f1-mediated cleavage activity, especially for Os-sg1.1, which contained A-U substituted to C-G at the stem 1 region of the tracrRNA and showed 1.56-fold increasement in GFP activation efficiency over WTOsCas12f1 (Fig. 2e). Thus, the Os-sg1.1 variant was selected for further optimization of OsCas12f1. Based on the first round optimization of OsCas12f1 sgRNA, we speculated that substitution with C-G base pair in sgRNA could be of benefit to increasing OsCas12f1 activity. To confirm this hypothesis, we substituted more base pairs with C-G base pair on Os-sg1.1, creating a sgRNA library with 13 variants. Through the second round sgRNA screen, we identified 8 sgRNA variants showing higher activity than that of Os-sg1.1. Among these 8 sgRNA variants, Os-sg2.6 outperformed over other variants (Fig. 2f).

We then sought to determine whether the respective increases in OsCas12f1 activity through protein and sgRNA engineering were additive effects, we first used the Os-sg1.1 sgRNA variant to guide the OsCas12f1-D52R protein variant. This combined variant showed higher cleavage activity than either variant alone (Supplementary Fig. 8b). Os-sg2.6 was then used to guide OsCas12f1-D52R, which outperformed

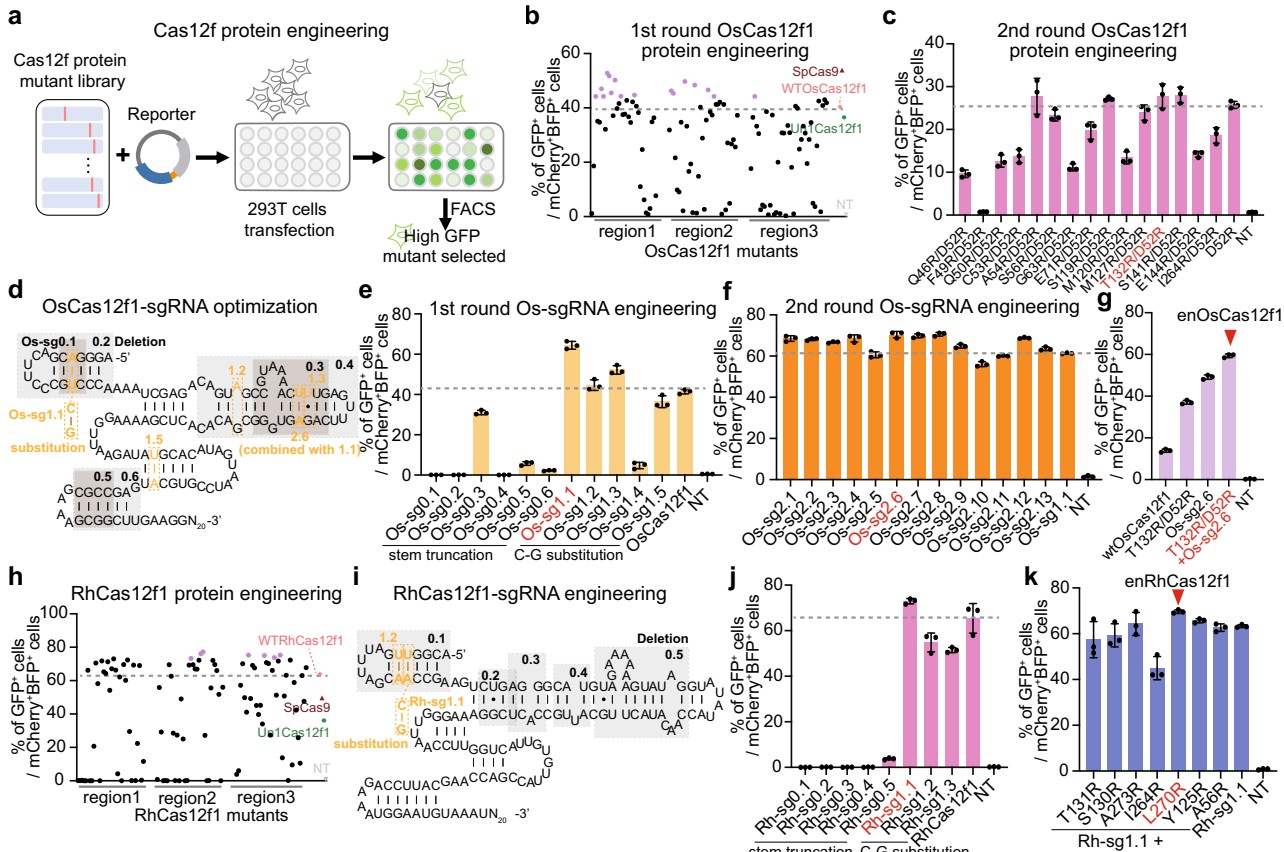

**Fig. 2 | Rational protein engineering and sgRNA optimization for high-efficiency Cas12f1. a** Scheme of protein engineering strategy. Mutants showing higher EGFP activation were selected for further optimization. **b** The first round high-efficiency mutant screen of OsCas12f1. The wild-type OsCas12f1 (WTOs-Cas12f1), Un1Cas12f1_ge4.1, SpCas9, and the mutant selected for next round screen are indicated. Target sequence (PAM-20nt protospacer): TTTC-CCATTACAGTAGGAGCATAC. **c** Second round enhanced OsCas12f1 variants screen by combining D52R with other arginine substitution mutants. Target sequence (PAM −20nt protospacer): TTTC-tttccctagggtccagcttcaaat. Values and error bars represent mean and s.d. (n = 3). **d** Engineering strategy for stem-loop deletion of OsCas12f1 sgRNA. **e** sgRNA engineering for OsCas12f1. The optimal sgRNA (Os-sg1.1), which was chosen for further engineering, is marked in red. Target sequence (PAM −20nt protospacer): TTTC-CCATTACAGTAGGAGCATAC. Values and error bars represent mean and s.d. (n = 3). **f** Second round sgRNA engineering by

including C-G base pair substitution on Os-sg1.1. Target sequence (PAM-20nt protospacer): TTTC-CCATTACAGTAGGAGCATAC. Values and error bars represent mean and s.d. (n = 3). **g** Increased EGFP activation efficiency by combining Cas12f1 protein mutant (T132R/D52R) and sgRNA variant (Os-sg2.6). Target sequence (PAM-20nt protospacer): TTTC-tttccctagggtccagcttcaaat. enOsCas12f1 is indicated with red triangle. Values and error bars represent mean and s.d. (n = 3). **h** Enhanced mutant screen of RhCas12f1. Each dot indicates one mutant. **i, j** Optimization of RhCas12f1 sgRNA to increase activity. The marked sgRNA (Rh-sg1.1) was selected for further optimization. Values and error bars represent mean and s.d. (n = 3). **k** Combination of Rh-sg1.1 variant with protein variant further increased the EGFP activation efficiency of Rh-sg1.1 variant. The best combination was indicated as enRhCas12f1 that is marked with red triangle. Values and error bars represent mean and s.d. (n = 3). Source data are provided as a Source Data file.

over D52R/Os-sg1.1 combination variant (Supplementary Fig. 8c–f). Lastly, by combining T132R with D52R/Os-sg2.6, generating the most efficient combination variant, named as enOsCas12f1 (Fig. 2g and Supplementary Fig. 8g). The enOsCas12f1 exhibited 9.4-fold increasement than that of WTOsCas12f1 at *DMD* locus (Supplementary Fig. 8h).

For generating enRhCas12f1, seven high-performance protein variants were chosen for combination with the most efficient sgRNA variants, Rh-sg1.1 (Fig. 2h–j). Among these combination variants, L270R + Rh-sg1.1 combination variant outperformed over others, showing 1.61-fold improvement over WTRhCas12f1 at endogenous *PCSK9* locus (Fig. 2k and Supplementary Fig. 8i).

In addition, the in vitro PAM characterization assay was performed to determine the PAM preference of engineered Cas12f1, indicating the enOsCas12f1 preferred 5′-TTH (H = not G) > 5′-TTG, while enRhCas12f1 recognized PAM as 5′-CCD (Supplementary Fig. 9). Based on the in vitro PAM characterization result, we then further compared the PAM preferences of different Cas12f1s, including OsCas12f1, enOsCas12f1, and Un1Cas12f1_ge4.1, in HEK293T cells using GFP activation reporter with fixed T at position −2 and −3 of 5′-PAM (5′-NTTN).

The GFP activation results suggested that enOsCas12f1, recognized PAM as 5′-TTN, showing a broader target range than that of WTOs-Cas12f1 and Un1Cas12f1_ge4.1, respectively preferred 5′-YTTH and 5′-TTTR (R = A or G) (Fig. 3a). The reporter with fixed C at position −2 and −3 of 5′-PAM (5′-NCCN) was used for RhCas12f1 and enRhCas12f1. The efficiency of enRhCas12f1 at all of 5′-CCN PAM sites was improved compared to that of WTRhCas12f1 (Fig. 3b). Additionally, the indel frequency analysis at 44 endogenous loci further confirmed that enOsCas12f1 was active at 5′-NTTN target sites with >10% indel at 5′-TTC (12 out of 12 sites), 5′-TTA (7 out of 9 sites), 5′-TTT (9 out of 11 sites) and 5′-TTG (4 out of 11 sites), indicating the PAM preferences of enOsCas12f1 as 5′-TTC > 5′-TTA > 5′-TTT > 5′-TTG (Fig. 3c, e). As expected, Un1Cas12f1_ge4.1 induced indels predominately at the 5′-TTTR sites, showing >10% indel at 5′-TTA (4 out of 9 sites) and 5′-TTG (2 out of 11 sites) (Fig. 3c, e). The PAM preference of enRhCas12f1 was also analyzed by evaluating the indel frequency at 45 endogenous loci, revealing that enRhCas12f1 achieved >10% indel at 5′-CCA (9 out of 12 sites), 5′-CCT (4 out of 11 sites) and 5′-CCG (3 out of 11 sites), suggesting enRhCas12f1 recognized 5′-CCD PAM (Fig. 3d, f).

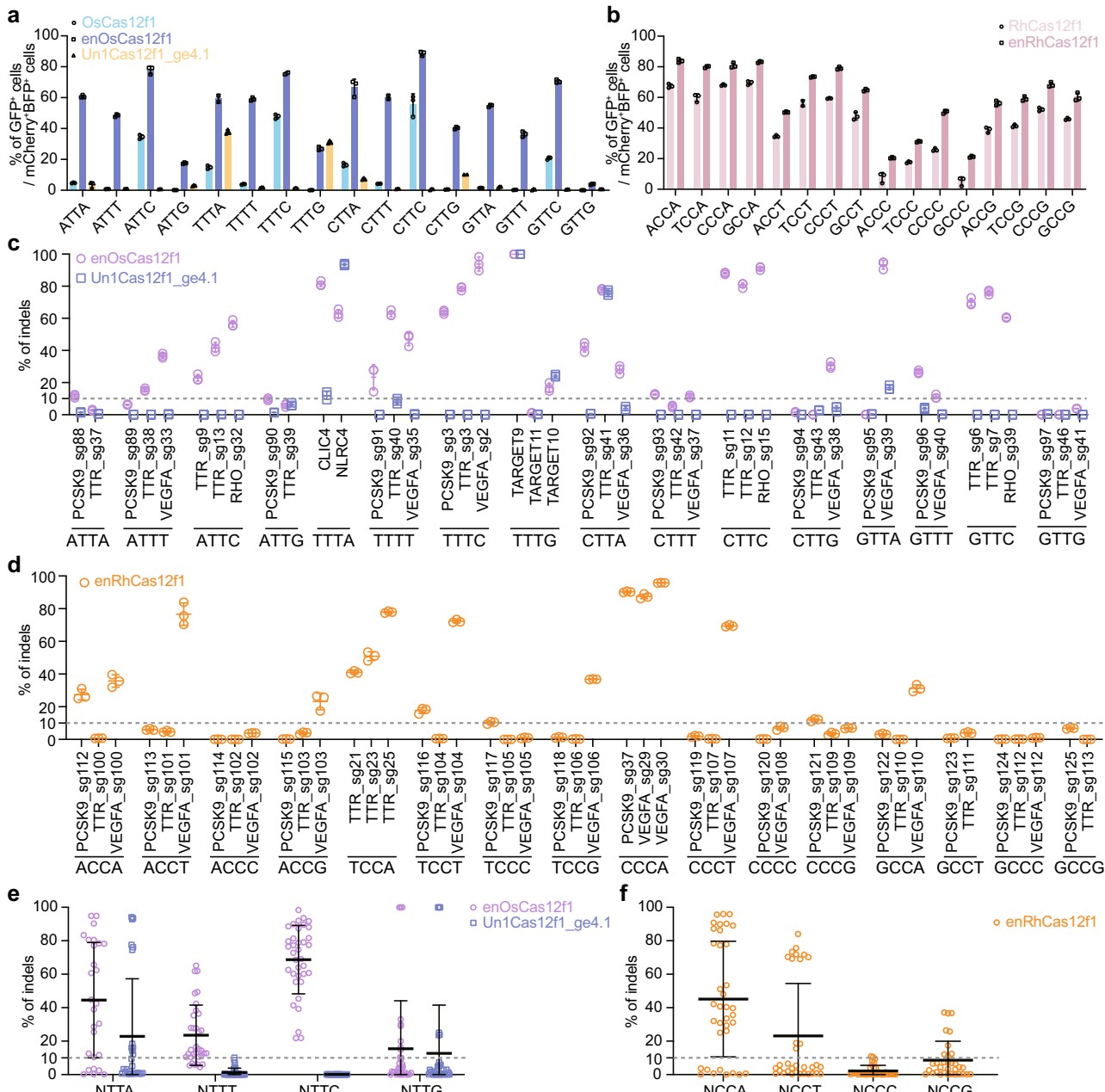

**Fig. 3 | PAM preferences of enOsCas12f1 and enRhCas12f1. a** PAM preferences of OsCas12f1, enOsCas12f1 and Un1Cas12f1_ge4.1 analyzed by GFP activation system. Target sequence: CCATTACAGTAGGAGCATAC. Values and error bars represent mean and s.d. ($n$ = 3). **b** Comparison of RhCas12f1- and enRhCas12f1-preffered PAM. Target sequence: CCATTACAGTAGGAGCATAC. Values and error bars represent mean and s.d. ($n$ = 3). **c**, **d** Validation the PAM preferences of enOsCas12f,

enRhCas12f1 and Un1Cas12f1_ge4.1 at endogenous loci. Values and error bars represent mean and s.d. ($n$ = 3). **e**, **f** Summary of indel efficiencies of enOsCas12f1, Un1Cas12f1_ge4.1, and enRhCas12f1. Values and error bars represent mean and s.d. from biologically independent experiments. Source data are provided as a Source Data file.

Therefore, protein engineering, which may increase the binding ability of Cas12f to nucleic acids, combining C-G base pair substitution in sgRNA can improve the cleavage activity of OsCas12f1 and RhCas12f1, and broaden the target range of OsCas12f1.

### enOsCas12f1 and enRhCas12f1 enable robust genomic editing in human cells

We next asked whether enOsCas12f1 and enRhCas12f1 could efficiently edit endogenous genomic loci in human cells. To comprehensively compare the editing efficiencies of enOsCas12f1, enRhCas12f1, and the published high-performance Cas12f1, Un1Cas12f1_ge4.1, we quantified targeting of all access sites in the

exons of *PCSK9*, *TTR*, and *VEGFA*, based strictly on PAM sequence without consideration for potential sgRNA and target feature contributing toward Cas nuclease activity, such as GC content[26,27]. In total, the indel frequency was quantified at 30 sites targeted by enOsCas12f1 (5'-NTTC PAM), 61 sites targeted by enRhCas12f1 (5'-TCCA and 5'-CCCA PAM), and 27 sites targeted by Un1Cas12f1_ge4.1 (5'-TTTR PAM). The results showed that enOsCas12f1 induced indels (>1%) in all of 30 tested sites with a maximal efficiency of 96.2%, while enRhCas12f1 induced indels (>1%) in 53 of the 61 tested loci with a maximal efficiency of 93.3%. By contrast, Un1Cas12f1_ge4.1 generated relatively lower indel frequencies (>1%) in 22 sites across 27 tested loci, with a maximal efficiency of 60.6% (Fig. 4a).

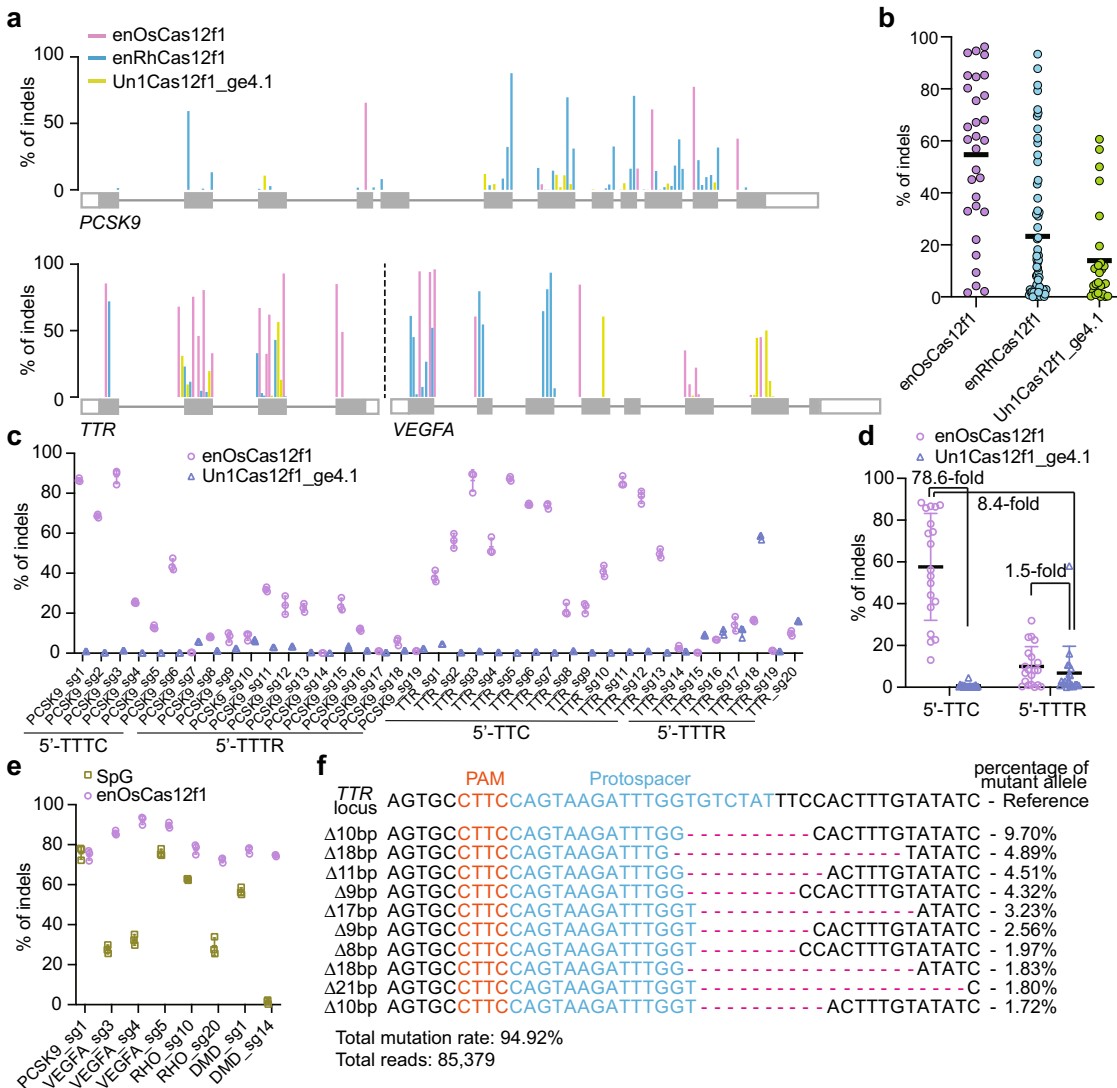

**Fig. 4 | Comprehensive validation of genomic editing efficiency of enOsCas12f1 and enRhCas12f1 in human cells. a** Distribution of all exon-located target sites that are accessible for enOsCas12f1 (5′-NTTC PAM), enRhCas12f1 (5′-CCCA PAM), and Un1Cas12f1_ge4.1 (5′-TTTR PAM), and the indel frequencies are indicated by mean values of three replicates, determined by NGS. The exon (gray solid squares) is connected by intron (lines), and UTRs are shown as hollow boxes. **b** Indel frequencies of enOsCas12f1, enRhCas12f1 and Un1Cas12f1_ge4.1 at endogenous genomic loci. Each dot represents single target site, value means average of three replicates. Bars represent means. **c** Comparison of editing efficiencies of

enOsCas12f1 and Un1Cas12f1_ge4.1 targeted by same sgRNAs at *PCSK9* and *TTR* loci. Values and error bars represent mean and s.d. (*n* = 3). **d** Average indel frequency of enOsCas12f1 and Un1Cas12f1_ge4.1 at 5′-TTC PAM and 5′-TTTR PAM target sites. Each dot represents single target site, value means average of 3 replicates. Error bars represent mean and s.d. **e** Comparison of editing efficiencies of enOsCas12f1 and SpG targeted by same sgRNAs at endogenous target loci. Values and error bars represent mean and s.d. (*n* = 3). **f** The distribution of mutant alleles by enOsCas12f1-mediated disruption at TTR locus, the top 10 mutant alleles are represented. Source data are provided as a Source Data file.

On average, enOsCas12f1 (54.7 ± 29.8%, mean ± s.d.) thus exhibited 3.9-fold higher efficiency and enRhCas12f1 (23.3 ± 26.8%) showed 1.7-fold higher efficiency than Un1Cas12f1_ge4.1 (14.0 ± 18.1%), respectively (Fig. 4b). When assessed the indel frequency induced by enOsCas12f1 and Un1Cas12f1_ge4.1 targeted by exactly same sgRNAs at *PCSK9* and *TTR* loci, we found that enOsCas12f1 showed 78.6-fold higher indel frequency on average at 5′-TTC PAM sites than that of Un1Cas12f1_ge4.1, and 8.4-fold higher efficiency at their own preferred 5′-PAM (5′-TTC for enOsCas12f1 and 5′-TTTR for Un1Cas12f1_ge4.1) (Fig. 4c, d). According to these advantages of enOsCas12f1 over Un1Cas12f1_ge4.1, we achieved up to 54.4 ± 29.9% and 59.1 ± 23.1% editing efficiency at therapeutic target loci *PCSK9* and *TTR* respectively, while Un1Cas12f1_ge4.1 showed relatively lower editing efficiency, with an average efficiency of 2.3 ± 1.9% and 15.2 ± 18.7% (Fig. 4c, d). Additionally, the activity of enOsCas12f1 and SpG[28] was compared by indel analysis at endogenous sites of 5′-TTC-

N$_{20}$−3′-NGN at *PCKS9*, *VEGFA*, *RHO*, and *DMD* loci, indicating that enOsCas12f1 outperformed SpG at these target sites (Fig. 4e).

High throughput sequencing of target loci revealed that both enOsCas12f1 and enRhCas12f1 predominantly generated deletions that altered the protospacer sequences rather than insertions (Fig. 4f, Supplementary Fig. 10). The center of the deletion position was located at the PAM-distal region outside of the protospacer sequences (Fig. 4f, Supplementary Fig. 10), which was similar to that of Un1Cas12f1 and AsCas12f1[14,15].

### The specificities of enOsCas12f1- and enRhCas12f1-mediated genome editing
We first evaluated the mismatched tolerances of enOsCas12f1 and enRhCas12f1 by tilling single or adjacent two mismatches in spacer sequences. For the *PCSK9* locus, enOsCas12f1 did not tolerate single mismatch at position 3/5/11, while mismatch at other positions slightly

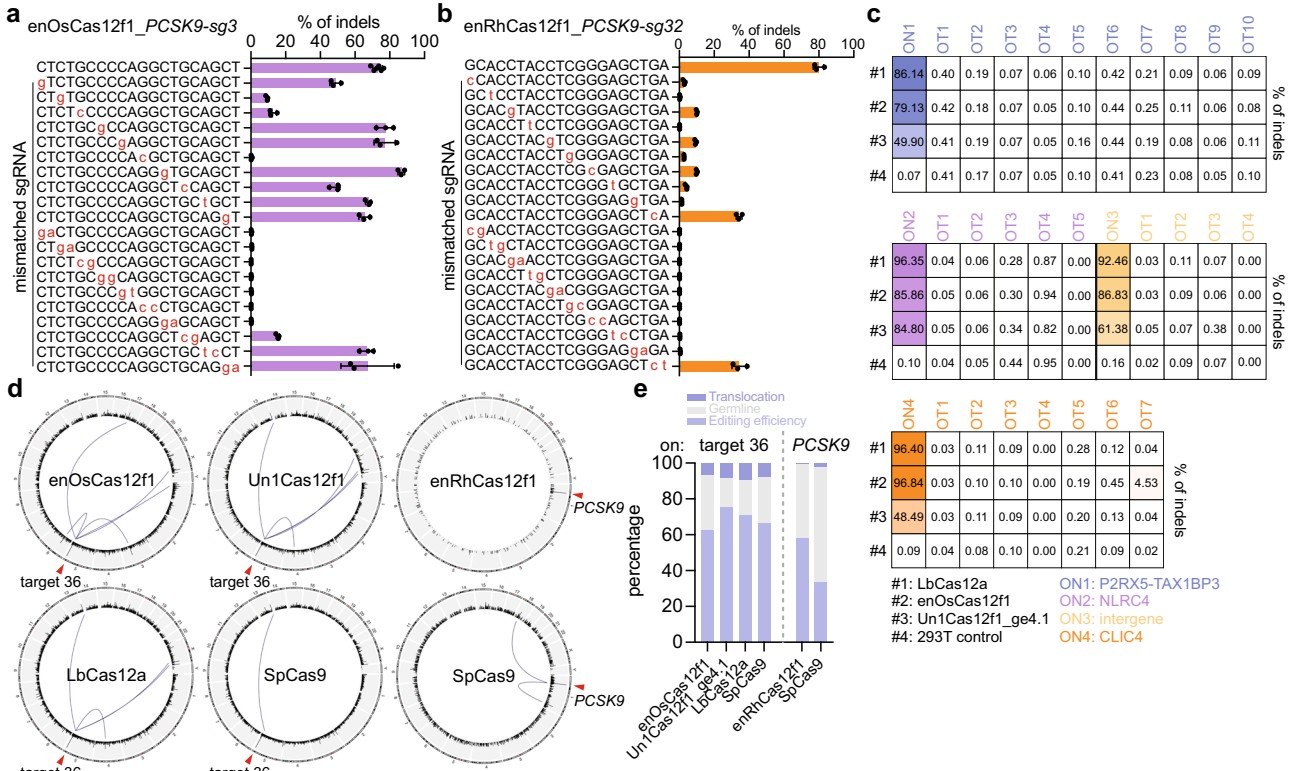

**Fig. 5 | Specificities of enOsCas12f1- and enRhCas12f1-mediated genome editing in human cells. a** Effects of mismatched sgRNA with 1 bp or 2 bp on activities of enOsCas12f1 at *PCSK9* locus. Values and error bars represent mean and s.d. (*n* = 3). **b** Mismatch tolerance of enRhCas12f1 at *PCSK9-sg32*. Values and error bars represent mean and s.d. (*n* = 3). **c** Off-target efficiency of LbCas12a, enOsCas12f1, and Un1Cas12f1_ge4.1 at in silico predicted off-target sites, determined by targeted deep sequencing. **d, e** PEM-seq genome-widely quantified the translocation efficiencies induced by off-target indels of enOsCas12f1 and enRhCas12f1. Circos plot showing the off-target sites that were linked to the bait DSB (red triangle, **d**). Percentages of translocation, germline, and editing efficiency calculated by PEM-seq analysis of enOsCas12f1, Un1Cas12f1_ge4.1, LbCas12a, SpCas9, and enRhCas12f1 (**e**). Source data are provided as a Source Data file.

reduced enOsCas12f1-mediated editing efficiency (Fig. 5a), which was validated by GFP activation system (Supplementary Fig. 11). However, adjacent two mismatch at position 1–16 substantially reduced enOs-Cas12f1 activity (Fig. 5a). The mismatched tolerance of RhCas12f1 were assessed at endogenous *PCSK9* locus and by GFP activation reporter system, indicating that enRhCas12f1 partially tolerate base pair mismatches at PAM-distal region especially at position 19 and 20, while mismatches close to PAM could substantially reduce the activity of enRhCas12f1 (Fig. 5b and Supplementary Fig. 11).

We then performed targeted deep sequencing at the in-silico predicted off-target sites (*P2RX5-TAX1BP3*, an intergenic region, *NLRC4* and *CLIC4*)[15]. Targeted deep sequencing indicated that the on-target editing efficiency of enOsCas12f1 was comparable to that of LbCas12a, and slightly higher than that of Un1Cas12f1_ge4.1. Similar to LbCas12a and Un1Cas12f1_ge4.1, enOsCas12f1 also showed strikingly low off-target effects at the potential off-target sites, while a low off-target effect found at CLIC4 OT7 site for enOs-Cas12f1 (Fig. 5c).

Finally, PEM-seq[29] was performed to quantify the genome-wide editing specificities of enOsCas12f1 and enRhCas12f1. When targeted at target 36 site, five off-target sites were found induced by enOsCas12f1 and Un1Cas12f1_ge4.1, four and one of off-target sites were found for LbCas12a and SpCas9, respectively (Fig. 5d). enOsCas12f1 exhibited 7.03% of translocation rate, which was comparable to that of Un1Cas12f1_ge4.1 (8.44%), LbCas12a (9.22%), and SpCas9 (8.19%) when targeted at target 36 site (Fig. 5e). enRhCas12f1 showed no detectable off-target site with low translocation efficiency for targeting *PCSK9* locus, while 2 off-target sites were found for SpCas9 (Fig. 5d, e). Together, these results suggested that enOsCas12f1 and enRhCas12f1

exhibited high genomic editing efficiency with a wide target range and low off-target effects.

### enOsCas12f1-mediated in vivo genome editing by single AAV delivery and enOsCas12f1-based epigenome editing and gene activation

The considerably small size of enOsCas12f1 suggested that its expression cassette could be packaged with multiple sgRNAs in a single rAAV vector, which could enable its therapeutic application to treat genetic disorders that require large fragment deletions, such as Duchenne muscular dystrophy (DMD)[30,31]. Previous work has shown that skipping exon 51 of the dystrophin gene can restore the disrupted open reading frame in as many as 13% of DMD patients carrying exon deletions[32]. To test whether enOsCas12f1 could be harnessed for DMD exon 51 deletion, we firstly screened efficient sgRNAs flanking exon 51 (5'sgRNA and 3'sgRNA), indicated enOsCas12f1 efficiently induced indels, while enRhCas12f1 and Un1Cas12f1_ge4.1 exhibited low editing efficiency at four of target sites (Supplementary Fig. 12a). We then combined efficient 5'sgRNA with 3'sgRNA to target enOsCas12f1 to DMD exon 51 in HEK293T cells (Fig. 6a). PCR-based assays revealed robust genomic deletion of exon 51 (~1700bp deletion) by enOsCas12f1 targeted by sg1 + sg16, which was more efficient than that of SpCas9 (~850 bp deletion), although the indel frequency of individual sgRNA of enOs-Cas12f1 was lower than that of SpCas9 (Fig. 6b and Supplementary Fig. 12a).

Precisely controlling of enOsCas12f1 activity across multiple dimensions such as dose and timing could undoubtedly reduce the potential toxicity and off-target effects induced by enOsCas12f1, especially for in vivo scenario where enOsCas12f1 is constitutively

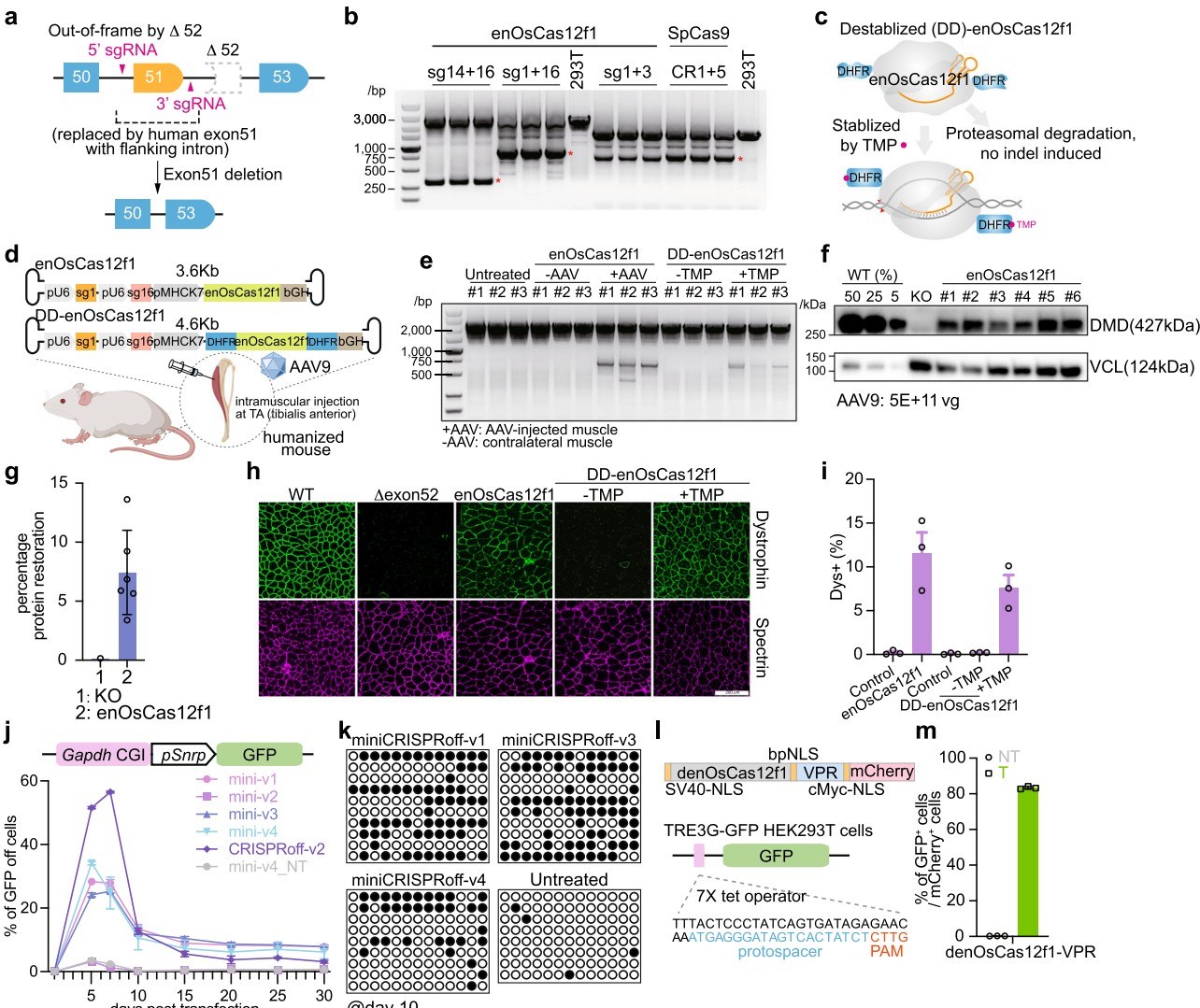

**Fig. 6 | Tunable enOsCas12f1-mediated in vitro and in vivo deletion of human DMD exon 51 and engineering enOsCas12f1 for epigenome editing and gene activation. a** Strategy for generating humanized DMD mutation mouse with human exon 51 replacement and exon 52 deletion. By deleting exon 51 can restore the dystrophin expression. Two sgRNAs that located before (5' sgRNA) and after (3' sgRNA) exon 51 are designed to delete exon 51. **b** enOsCas12f1- and SpCas9-mediated deletion of DMD exon 51 by paired sgRNAs in HEK293T cells. The exon 51 deleted bands were marked by red asterisk. This experiment was repeated two times showing similar results. **c** Scheme representing the strategy for destabilized enOsCas12f1 (DD-enOsCas12f1). **d** Overview of intramuscular injection of single AAV9 system in humanized mouse. **e** The in vivo editing efficiencies of enOsCas12f1 and DD-enOsCas12f1 were tested by genomic PCR. This experiment was repeated two times showing similar results. **f** Western blotting for detecting recovery of dystrophin (DMD) by enOsCas12f1 and DD-enOsCas12f1 in DMD model mouses.

Vinculin (VCL) protein level was used as internal control. **g** Percentage of recovered dystrophin by western blotting analysis. Values and error bars represent mean and s.d. ($n = 1$ for KO, $n = 6$ for enOsCas12f1). **h** DMD immunofluorescence staining. **i** Percentage of dystrophin positive fibers in enOsCas12f1 and DD-enOsCas12f1 treated muscles. Values and error bars represent mean and s.d. ($n = 3$). **j** GFP silencing activity of miniCRISPRoff-v1-v4 and CRISPRoff-v2. The stably GFP expressing HEK293T cells generated by *piggyBac* system were used. Values and error bars represent mean and s.d. ($n = 3$). **k** DNA methylation level on the *Snrp* promoter region. **l** Design strategy for denOsCas12f1-VPR adopted from Xu et al. The TRE3G-GFP reporter cell line was created by piggyBac system in HEK293T cells. **m** GFP activation efficiencies of denOsCas12f1-VRP. sgRNA congaing random spacer sequence served as non-target (NT) control. Values and error bars represent mean and s.d. ($n = 3$). Source data are provided as a Source Data file.

expressed via AAV delivery. To achieve precisely controlled enOsCas12f1, we fused enOsCas12f1 with the destabilized domains (DD) of *E. coli* dihydrofolate reductase (ecDHFR). The newly synthesized DD-enOsCas12f1 proteins are rapidly targeted for proteasomal degradation, which can be blocked by the small molecule trimethoprim (TMP)[33,34] (Fig. 6c).

To assess the in vivo deletion efficiencies of DMD exon 51 induced by enOsCas12f1 and DD-enOsCas12f1, we created a mouse model of DMD with exon 52 deletion and exon 51 replaced by human exon 51 with flanking intron sequences (Fig. 6a). Deletion of exon 52 prematurely terminates protein production of dystrophin, which can

be restored by removal of exon 51. We used AAV serotype 9 (AAV9) for local delivery of enOsCas12f1 / DD-enOsCas12f1 as well as its sgRNA expression cassette to skeletal muscle (Fig. 6d). Because of the single AAV-packageable size of CRISPR-OsCas12f1 system, we injected AAV9s into the tibialis anterior muscle with lower titer than that of SpCas9, which needs dual AAV due to its large size[35-37]. PCR-based detection across the genomic locus indicated the expected ~1700bp deletion (Fig. 6e). RT-PCR of mRNA extracted from whole muscle showed the transcripts with exon 51 deletion at efficiency of $22.7 \pm 9.2\%$ (mean $\pm$ s.d.) for enOsCas121, while $15.0 \pm 7.0\%$ for DD-OsCas12f1 (Supplementary Fig. 12b-d). western blotting of whole

muscle and immunostaining results further confirmed that the protein production of dystrophin was rescued by enOsCas12f1 and DD-enOsCas12f1 (Fig. 6f–h). Restoration of dystrophin protein occurred in 11.6 ± 4.0% and 7.6 ± 2.4% of myofibers treated by enOsCas12f1 and DD-enOsCas12f1, respectively (Fig. 6i).

We next tested the efficiency of enOsCas12f1-mediated epigenome editing, which was named miniCRISPRoff (1444 aa), by adopting the strategy of CRISPRoff, with protein size at 2,361 aa[11]. Four version of miniCRISPRoff were generated (Supplementary Fig. 13), among which miniCRISPRoff-v1, v3, and v4 silenced GFP in the *GAPDH-Snrp*-GFP stably expressed HEK293T cells (Fig. 6j and Supplementary Fig. 14a). Bisulfite sequencing indicated that the *Snrp* promoter was highly methylated by treatment with miniCRISPRoffs (Fig. 6k). Finally, the ability of enOsCas12f1-mediated gene activation was assessed by fusing enOsCas12f1 with VPR[14], which showed a robust gene activation of GFP in TRE3G-GFP HEK293T cells (Fig. 6l, m and Supplementary Fig. 14b). Totally, these results indicated that enOsCas12f1 can be engineered as versatile genome and epigenome editors.

## Discussion

Although compact Cas12f orthologs have been tested in the genome editing delivered by a single AAV vector in human cells, their relatively low editing efficiency and restricted PAM requirement may constrained their further application. Here, we characterize a set of type V-F CRISPR-Cas12f1 subfamily members from bacteria, and identify six that are functional in human cells (Fig. 1). By protein engineering combining with sgRNA optimization, we obtained enOsCas12f1 and enRhCas12f1 (Fig. 2), showed significantly higher genomic editing efficiency and a broader targeting range than that of Un1Cas12f1_ge4.1, which is the most efficient Cas12f reported to date and is comparable with SpCas9[15] (Figs. 3 and 4). The discovery of enOsCas12f1 and enRhCas12f1 greatly expanded the target range of Cas12f systems. Un1Cas12f1_ge4.1 required 5′-TTTR (R = A or G) PAM, while enOsCas12f1 was active at 5′-NTTN containing loci. Thus, enOsCas12f1 broadened the target range as much as 8-fold over that of Un1Cas12f1_ge4.1. The 5′-NCCD PAM of enRhCas12f1 is also a promising compensation for the 5′-T-rich PAM constrain of enOsCas12f1 and Un1Cas12f1_ge4.1 (Fig. 3).

Rational protein engineering combined with sgRNA optimization, which enable enhanced interaction of Cas protein with nucleic acids or sgRNA, and increased sgRNA stability, has been previously applied to enhance the editing activities, targeting range, or fidelity of Cas9 and Cas12 proteins[14,15,21,38,39]. This strategy was further validated in the current study. It is worth to note that the efficiencies of both OsCas12f1 and RhCas12f1 were substantially improved by substituting the A-U basepair in the first stem of sgRNA with G-C base pair (Fig. 2), which may be adopted for engineering other RNA-programmable DNA nuclease including IscB and TnpB.

enOsCas12f1 enables robust and specific genomic editing in vitro and in vivo, and can be applied for efficient deletion of large fragment in human genome, such as ~1700 bp deletion of exon 51 of dystrophin (Figs. 5 and 6). It has been shown that increased off-target mutations and DNA damage response could be triggered by constitutive nuclease activity of Cas proteins[40]. Acute manipulation of the activity of enOsCas12f1 within indicated time window and specific type of cells is a promising way to reduce these potential unexpected side effects. By conjugating the destabilized domains of ecDHFR to enOsCas12f1 (DD-enOsCas12f1), we achieved highly specific regulation of enOsCas12f1-mediated gene editing in vivo. It is worth mentioning that DD-enOsCas12f1 together with up to two sgRNAs could be packaged into a single AAV vector, that circumvents obstacles related to the larger size of Cas9/12 that cannot be packaged into a single AAV. Additionally, cell type specific promoters that usually contain longer sequences can be used for driving expression of enOsCas12f1 and DD-enOsCas12f1 to achieve more precise control of OsCas12f1 activity using systematic

delivery by AAVs, which is undoubtedly safer for therapeutic application.

It is also plausible that the hypercompact size of enOsCas12f1 (433 aa) and enRhCas12f1 (415 aa) could potentially enable their use in derivative genome engineering applications, including base editing, prime editing, retron editing, epigenome editing, and gene expression regulation[8,10–12,41]. Here, we engineered enOsCas12f1 for sufficient epigenome editing (miniCRISPRoff) and gene activation (enOsCas12f1-VPR). It is interesting to engineer miniCRISPRoff for more efficient and smaller size that can be packaged by single AAV in the future.

In summary, enOsCas12f1 and enRhCas12f1 represent high-performance gene editing tools with versatile applications, while temporally and spatially controlled DD-enOsCas12f1 is a promising platform for gene therapy.

## Methods

### Ethical statement

Our research complies with all relevant ethical regulations, and animal experiments have been approved by the Animal Care and Use Committee of Huidagene Therapeutics Co., Ltd, Shanghai, China.

### Computational analysis of CRISPR-Cas12f systems and PAM prediction

More than 200,000 bacteria genome were downloaded from NCBI database. Firstly, we used TBLASTN and UnCas12f protein to identify Cas12f-containing sequences of bacteria genomes downloaded from NCBI with *E* value<1e-10. Then, "0.Cas-Finder.pl" script was used to annotate the CRISPR array and Cas proteins of Cas12f-containing sequences. We further used "1.Cas12f-Finder.pl" to annotate the Cas12f proteins with conserve RuvC and Zn finger domain.

Then, the definition of the 5′ boundary of crRNA depends on the prediction of anti-repeat in tracrRNA. As we know, the direct repeats of mature Cas12s' crRNAs are generally in the 3′ end sequence of about 22 nt. Therefore, we used the 22 nt sequence at the 3′ end of DR to search the non-coding sequence between the Cas12f gene and CRISPR array.

We defined the non-coding sequence containing at least 9 A-U / C-G pairs, and at least 65% of A-U / C-G / G-U pairs with 22 nt sequence at the 3′ end of DR as the anti-repeat sequence. We further extended 150 nt upstream of anti-repeat to obtain potential tracrRNA sequences. Then, using RNAfold to predict the secondary structure of the potential tracrRNA sequences, we retain the sequences with conservative secondary structure in Cas12f family. Based on the above principles, we wrote "2. Cas12f. tracrRNA.Finder. pl" script to predict the tracrRNA sequences of Cas12f variants.

The scripts for Cas12f identification and tracrRNA prediction has been deposited on github.

We initially predicted the PAMs for 34 CRISPR-Cas12f1 systems by CRISPRTarget[42], ten of these CRISPR-Cas12f1 systems were successfully predicted (Supplementary Data 1). The PAMs of resting CRISPR-Cas12f1 systems were then predicted based on the protein homology with those Cas12f1s whose PAMs were successfully obtained by CRISPRTarget.

### Plasmids construction and purification of Cas12f1 proteins

Human codon-optimized Cas12f and sgRNA were synthesized and cloned to generate pCAG_NLS-Cas12f-NLS_pA_pU6_gRNA scaffold-2x BpiI_pCMV_mCherry_pA by NEBuilder (New England Biolabs). The spacer sequences were annealed and ligated to BpiI sites. All sequences are listed in Supplementary Note 1.

For the generation of Cas12f1 protein mutants, region 1-3 of OsCas12f1 and RhCas12f1 were divided into 11 segments containing 17 amino acid residues in length. Eleven backbone mutants for OsCas12f1 and RhCas12f1, respectively, were generated by replacing the above mentioned 11 segments with BpiI recognition sequence by PCR and Gibson assembly method using NEBuilder HiFi DNA Assembly Master

Mix (New England Biolabs). The specific mutation is then introduced by incorporation of annealed oligos containing mutation by BpiI digestion and T4 DNA ligase ligation (Supplementary Fig. 7).

The full length of OsCas12f1, enOsCas12f1, RhCas12f1, and enRhCas12f1 was cloned into pET-32a to express Cas12f1 proteins with C-terminal 6xHis. Plasmids were transformed into *Escherichia coli* BL21(DE3) cells and grown at 37 °C to $OD_{600}$ of 0.6 and then induced for protein expression by 0.5 mM IPTG incubated at 18 °C overnight. Cells were harvested and lysed by sonication in Buffer A (50 mM Tris-HCl (PH = 8.0), 50 mM imidazole, 1.5 M NaCl). After centrifugation, the supernatant was gatherd and loaded onto the HisTrap HP column (Cytiva) and eluted with Buffer B (50 mM Tris-HCl (PH = 8.0), 600 mM imidazole, 1.5 M NaCl). The eluted protein was exchanged into Buffer C containing 20 mM Tris-HCl (PH = 8.0), 0.3 M NaCl, 1 mM DTT, and 2% (v/v) glycerol. The protein was then loaded on a HiTrap Heparin HP column (Cytiva), equilibrated with Buffer C, and eluted using a linear gradient of increasing NaCl concentration from 0.3 M to 2.0 M. Obtained protein was stored in Buffer D (25 mM Tris-HCl (PH = 8.0), 150 mM NaCl, 2 mM DTT and 1 mM $MgCl_2$). For long-term storage, the protein was supplemented with 10% (v/v) glycerol, then flash-frozen in liquid nitrogen and stored at −80 °C.

## sgRNA synthesis
The sgRNAs were prepared by in vitro transcription using a MEGA shortscript T7 kit (Life Technologies) and purified by a MEGA clear kit (Life Technologies). DNA templates for T7 transcription were generated by PCR using primers containing a T7 promoter. Sequences of these sgRNAs are provided in Supplementary Table 3.

## In vitro cleavage assay and PAM characterization
Cas12f1 ribonucleoprotein (RNP, 1 µM) complexes were assembled by mixing Cas12f1 protein with sgRNA at 1:1 molar ratio followed by incubation assembly buffer (10 mM Tris-HCl, pH 7.5, 100 mM NaCl, 1 mM EDTA, 1 mM DTT) at 37 °C for 30 min. Five nM of soercoiled or linear plasmids containing target sequences were incubated with 250 nM Cas12f1 RNP in reaction buffer (2.5 mM Tris–HCl, pH 7.5, 25 mM NaCl, 0.25 mM DTT, and 10 mM MgCl2) at 46 °C or indicated temperature for testing optimal temperature for one hour. The reaction was stopped with quenching buffer (20 mM EDTA, 0.1 mg/ml proteinase K). The digested product was analyzed with 1% of agrose gel. For run-off sequencing the digested product was purified and subjected to Sanger sequencing.

In vitro PAM characterization was performed as previously described[43]. Briefly, the dsDNA library with 7-bp random sequences followed by protospacer seqcuence was created by PCR with primer with 7 N. The in vitro cleavage was performed as above mentioned. The cleaved product with 7 N sequence was gel purified, adapter ligated and PCR for NGS. The top 1000 enriched PAM sequences were used to draw PAM motifs by WebLogo.

## Size-exclusion chromatography
To validate the Cas12f1–sgRNA complex formation, Cas12f1 RNP was assembled in vitro with 4:3 molar ratio of protein:sgRNA in buffer D at 37 °C for 30 min and analyzed on Superdex 200 Increase 10/300 column (Cytiva), equilibrated with Buffer D. Buffer E (20 mM Tris-HCl (PH = 8.0), 500 mM NaCl, 1 mM DTT and 5 mM $MgCl_2$) was used for analysis of Cas12f1 protein without sgRNA in view of the fact that OsCas12f1 protein could not be eluted from the column equilibrated with Buffer D, which may be due to non-specific interaction with the resin. The Gel Filtration Standard (Bio-Rad, # 1511901) was used for calibration.

## Cell culture, transfection, and flow cytometry analysis
HEK293T cells (Stem Cell Bank, Chinese Academy of Sciences) cultured in DMEM supplemented with 10% FBS and penicillin/ streptomycin were seeded on 24-well poly-D-lysine coated plates (Corning). For EGFP activation assay, transfection was conducted following the manufacturer's manual with 3.2 µl of PEI (Polyscience) and 1.6 µg of plasmids (0.8 µg of reporter plasmids + 0.8 µg of Cas12f expressing plasmids). Forty-eight hours after transfection, flow cytometry analysis was performed to evaluate the EGFP activation efficiency. For analyzing the indel efficiency of endogenous gene, HEK293T cells were transfected with 2 µl of PEI and 1 µg of plasmids expressing Cas12f and sgRNA cassette. The mCherry-positive cells were collected by FACS sorting at 72 h after transfection.

## Indel efficiency analysis at human endogenous genomic loci
Eight thousand sorted cells were harvested for genomic DNA extraction by addition of 20 µl of lysis buffer (Vazyme) following the manufacturer's manual. For TIDER test, the genomic region in the vicinity of Cas nuclease target site was amplified by Phanta Max Super-Fidelity DNA Polymerase (Vazyme) using nested PCR. Purified PCR products were Sanger sequenced and analyzed as previously described[44]. For deep sequencing analysis, the targeted genomic region was amplified by Phanta Max Super-Fidelity DNA Polymerase (Vazyme) using nested PCR, primers with barcode were used. PCR products were purified by Gel extraction kit (Vazyme) and sequenced on an Illumina HiSeq X System (150-bp paired-end reads). Forward reads were aligned to the reference sequences using BWA (v0.7.17-r1188) with parameter of "bwa mem -A2 -O3 -E1". At each target, editing was calculated as the percentage of total reads containing desired edits without indels within a 10-bp window of the cut site. The target site informations are provided in Supplementary Table 2.

## PEM-seq analysis
PEM-seq in HEK293 cells was performed as previously described[29]. Briefly, expression plasmids for enOsCas12f1, LbCas12a, SpCas9, and Un1Cas12f1_ge4.1 targeted at target 36, as well as enRhCas12f1 and SpCas9 targeted at *PCSK9* were transfected into HEK293 cells by PEI respectively, and after 72 h, positive cells were harvested for DNA extraction. The 20 µg genomic DNA was fragmented with a peak length of 300-700 bp by Covaris sonication. DNA fragments were tagged with biotin by a one-round biotinylated primer extension at 5′-end, and then primer removal by AMPure XP beads and purified by streptavidin beads. The single-stranded DNA on streptavidin beads is ligased with a bridge adapter containing 14-bp RMB, and PCR product was performed nested PCR for enriching DNA fragment containing the bait DSB and tagged with illumine adapter sequences. The prepared sequencing library was sequenced on a Hiseq 2500, with a 2 × 150 bp.

## Animals
All animal experiments were performed and approved by the Animal Care and Use Committee of Huidagene Therapeutics Co., Ltd, Shanghai, China. Mice were housed in a barrier facility with a 12-hour light/ dark cycle and 18–23 °C with 40–60% humidity. Diet and water were accessible at all times. DMD mice were generated in the C57BL/6 J background using the CRISPR-Cas9 system. Duchenne muscular dystrophy (DMD) is the most common sex-linked lethal disease in man, thus male mice were selected for this study.

## Intramuscular injection
AAVs were produced PackGene Biotech (Guangzhou, China), and applied iodixanol density gradient centrifugation for purification. For intramuscular injection, DMD mice were anesthetized, and TA (tibialis anterior) muscle was injected with 50 µL of AAV9 ($5 \times 10^{11}$ vg) preparations or with same volume saline solution. After AAV9 intramuscular injection 3 weeks, mice were anesthetized, euthanized and TA (tibialis anterior) muscle was collection.

## RT-PCR and TA cloning

Muscles total mRNA was extracted and cDNA was synthesized using a HiScript II One Step RT-PCR Kit (Vazyme, P611-01) following the manufacturer's protocol. Then, each 20 μl PCR reaction contained approximately 2 μl cDNA, 0.25 μM of each forward and reverse primers, and 10 μl of Ex taq (Takara, RR001A) was performed on a C1000 Touch Thermal Cycler (Bio-Rad). Amplification conditions consisted of an initial hold for 5 min followed by 35 cycles of 95 °C for 30 s, 60 °C for 30 s, and 72 °C for 30 s. PCR products were analyzed by gel electrophoresis.

For detected RNA splicing, TA cloning was performed according to the protocol of the pEASY-T5 Zero Cloning Kit (TransGen Biotech, CT501-01). Brief, PCR products were used agarose gel electrophoresis to verify the quality and quantity. 4 μl PCR products and pEASY-T5 Zero Cloning vector were gently mixed well, incubate at room temperature for 10 minutes, and then add the ligated products to 50 μl of Trams 1-T1 phage resistant chemically competent cell and plated on LB/Amp+, followed by sequencing with M13F.

## Western blot

Muscle samples were homogenized with RIPA buffer supplemented with protease inhibitor cocktail. Lysate supernatants were quantified with Pierce BCA protein assay kit (Thermo Fisher Scientific, 23225) and adjusted to an identical concentration using $H_2O$. Samples were mixed with in NuPAGE LDS sample buffer (Invitrogen, NP0007) and 10% β-mercaptoethanol followed by boiled at 70°C for 10 min. 20 μg total protein per lane was loaded into 3 to 8% tris-acetate gel (Invitrogen, EA03752BOX) and electrophoresed for 1 hours at 200 V. Protein was transferred on a PVDF membrane under the wet condition at 350 mA for 3.5 hours. The membrane was blocked in 5% non-fat milk in TBST buffer and then incubated with primary antibody labeling specific protein. After washing three times with TBST, the membrane was further incubated with HRP conjugated secondary antibody (1:1000 dilution, Beyotime, A0216) specific to the IgG of the species of primary antibody against dystrophin (1:1000 dilution, Sigma, D8168) and vinculin (1:1000 dilution, CST, 13901 S). The target proteins were visualized with Chemiluminescent substrates (Invitrogen, WP20005).

## Immunofluorescence

Tissues were collected and mounted in optimal cutting temperature (OCT) compound and snap-frozen in liquid nitrogen. Serial frozen cryosections (10 μm) were fixed for 2 hours in 37°C followed by permeabilized with PBS + 0.4%Triton-X for 30 min. After washing with PBS, samples were blocked with 10% goat serum for 1 hours at room temperature. Then, the slides were incubated overnight at 4°C with primary antibodies against dystrophin (1:100 dilution, Abcam, ab15277) and spectrin (1:500 dilution, Millipore, MAB1622). After that, samples were washed extensively PBS and incubated with compatible secondary antibodies (Alexa Fluor 488 AffiniPure donkey anti-rabbit IgG (1:1000 dilution, Jackson ImmunoResearch labs, 711-545-152) or Alexa Fluor 647 AffiniPure donkey anti-mouse IgG (1:1000 dilution, Jackson ImmunoResearch labs, 715-605-151)) and DAPI for 2 h at room temperature. Samples were washed for 10 min with PBS and repeated three times. And then, slides were sealed with fluoromount-G mounting medium. All images were visualized under Nikon C2. The amount of dystrophin-positive muscle fibers is represented as a percentage of total spectrin-positive muscle fibers.

## Efficiency detection on miniCRISPRoff

One microgram of mCherry containing plasmids expressing miniCRISPRoffs and CRIPSRoff were transfected into Snrp-GFP stably expressed HEK293T cells. Two days after transfection, mCherry-positive cells were sorted and cultured for FACS analysis at the indicated time.

For bisulfite sequencing analysis, genomic DNA was treated by BisulFlash DNA Modification Kit (EPIGENTEK) as the manufacturer's protocols. PCR amplicon of GAPDH-Snrp promoter was purified and cloned into TA cloning vector (VAYZYME). Colonies were randomly picked for Sanger sequencing. Primers were provided in Supplementary Table 3.

## Statistics and reproducibility

Frequency, mean, and standard deviations were calculated using GraphPad Prism 8. Whole-genome sequencing analysis was conducted using BWA (v0.7.17-r1188) with parameter of "bwa mem -A2 -O3 -E1". PEM-seq data analysis was performed using PEM-Q pipeline with default parameters. Two or three biologically independent replicates were performed, which was demonstrated in the figure legend. In this study, no statistical method was used to predetermine sample size, and no data were excluded from the analyses. The experiments were not randomized and the Investigators were not blinded to allocation during experiments and outcome assessment.

## Reporting summary

Further information on research design is available in the Nature Portfolio Reporting Summary linked to this article.

## Data availability

Next-generation sequencing data of PEM-seq have been deposited at the Sequence Read Archive: PRJNA895582. Source data are provided with this paper.

## Code availability

The scripts for Cas12f identification and tracrRNA prediction have been deposited on github.

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

## Acknowledgements

This work was supported by HUIEDIT Therapeutics Co., Ltd. and HUIDAGENE Therapeutics Co., Ltd.

## Author contributions

Xiangfeng Kong, and Y.Z. conceived and designed the project. Xiangfeng Kong, and Xuqiang Kong performed most experiments with the help of L.W., Z.W., M.X., J.Z., X.S., Yinghui Wei, and N.Z. on plasmids construction; H.Z., and W.Z. on PEM-seq analysis; G.L., and J.L. on animal experiments; Yao Wang on efficiency test in human cells; W.B. on AAV production. Y.Y. provided legal support. L.S. provided advices. Y.Z. performed bioinformatic analysis. Xiangfeng Kong, Y.Z., and H.Y. co-supervised the project. Xiangfeng Kong, Y.Z., and H.Y. wrote and revised the manuscript.

## Competing interests

H.Y. and L.S. are cofounders of HuidaGene Therapeutics Co., Ltd. ("HuidaGene"). H.Y. and Y.Z. are cofounders of HuiEdit Therapeutics Co., Ltd. ("HuiEdit"). Provisional patent applications PCT/CN2022/089053 and PCT/CN2022/142467 have been filed with inventors X.K., X.Q.K., and L.W. and owned by the patent applicant HuidaGene, claiming especially the CRISPR-Cas12 systems herein. The remaining authors declare no competing interests.
