## [Peer Review File · Nature Communications]

Reviewers' Comments:

Reviewer #3:

Remarks to the Author:

Most of my reviews have been dealt with appropriately. However, the authors should be able to do an experimental analysis of the multimeric state of the reported Cas12f1 effectors. Comparison of the size-exclusion profiles of the two guide-less proteins is very confusing: how do the authors explain the elution of OsCas12f1 at 5 vol/ml, whereas RhCas12f1 elutes at 49 vol/ml ?? I strongly recommend that this experiment id repeated, with and without a sgRNA. Loading of the effector proteins with the sgRNA can either be done in vitro (mixing the purified protein with a synthetic guide), or by co-expression of the sgRNA and the protein in vivo.

Response to the editorial concern

Editorially, we also note that one of the reviewers from the previous round of review at Nature Biotechnology noted that your editing efficiencies in fig.4 were higher than expected, and we ask that you compare the Un1Cas12f1 protein used as a control with that of the value previously reported in the literature, and in case of differences, please explain.

Response: We appreciated your comments. In fact, we had tested the Un1Cas12f1_ge4.1 mediated editing efficiencies at four of target sites by exactly same spacer sequence that were previously reported in the literature (Kim DY et al 2022, Nat Biotechnol, DOI: 10.1038/s41587-021-01009-z). As shown in Fig. 5c in this paper, we found the indel frequencies induced by Un1Cas12f1_ge4.1 at *P2RX5-TAX1BP3*, *NLRC4*, *Intergene* and *CLIC4* were 49.90%, 84.80%, 61.38% and 48.49%, respectively, which are comparable to the value shown in Kim's paper with indel frequencies of 51.5%, 73.2%, 61.2% and 45.9% (Fig. 4b in Kim's paper).

These results confirmed that the Un1Cas12f1_ge4.1 used in this study was comparable to the literature one, although it showed low average efficiencies at the tested sites on *PCSK9*, *TTR* and *VEGFA* loci, which may be caused by the difference of target sites. Importantly, these results validated that our indel frequency analysis pipeline is reliable.

Point-by-point response to the reviewers' comments

Reviewer #3 (Remarks to the Author):

Most of my reviews have been dealt with appropriately. However, the authors should

be able to do an experimental analysis of the multimeric state of the reported Cas12f1 effectors. Comparison of the size-exclusion profiles of the two guide-less proteins is very confusing: how do the authors explain the elution of OsCas12f1 at 5 vol/ml, whereas RhCas12f1 elutes at 49 vol/ml ?? I strongly recommend that this experiment id repeated, with and without a sgRNA. Loading of the effector proteins with the sgRNA can either be done in vitro (mixing the purified protein with a synthetic guide), or by co-expression of the sgRNA and the protein in vivo.

Response: Thank you for your valuable suggestion. To validate the Cas12f1 – sgRNA complex formation, Cas12f1 RNP was assembled *in vitro* with 4 : 3 molar ratio of protein : sgRNA at 37 °C for 30 min and analyzed on Superdex 200 Increase 10/300 column (Cytiva). Size-exclusion chromatography results suggested that both OsCas12f1 and RhCas12f1 could form dimer in presence of sgRNA at least in the tested condition (Supplementary Fig. 5), which is similar to that of Un1Cas12f1 (Takeda SN et al 2021, Mol Cell, DOI: 10.1016/j.molcel.2020.11.035; Xiao R et al 2021, Nucleic Acids Res, DOI: 10.1093/nar/gkab179).

Supplementary Figure 5. OsCas12f1-sgRNA and RhCas12f1-sgRNA complex formation.

Reviewers' Comments:

Reviewer #3:

Remarks to the Author:

My remaining concern about the multimeric state of both Cas12f1 variants has been appropriately addressed by the additional gel filtration experiment. Hence, I fully support accepting this work for publication in Nature Communications. Congratulations with this impressive study